# Low Bone Mineral Density and Risk for Osteoporotic Fractures in Patients with Chronic Pancreatitis

**DOI:** 10.3390/nu13072386

**Published:** 2021-07-13

**Authors:** Miroslav Vujasinovic, Lorena Nezirevic Dobrijevic, Ebba Asplund, Wiktor Rutkowski, Ana Dugic, Mashroor Kahn, Ingrid Dahlman, Maria Sääf, Hannes Hagström, Johannes-Matthias Löhr

**Affiliations:** 1Department of Upper Abdominal Diseases, Karolinska University Hospital, 141 86 Stockholm, Sweden; wiktor.rutkowski@ki.se (W.R.); hannes.hagstrom@ki.se (H.H.); matthias.lohr@ki.se (J.-M.L.); 2Department of Medicine, Huddinge, Karolinska Institutet, 171 76 Stockholm, Sweden; lorena.dobrijevic@stud.ki.se (L.N.D.); ebba.asplund@stud.ki.se (E.A.); ana.dugic@ki.se (A.D.); mashroor.khan@stud.ki.se (M.K.); ingrid.dahlman@ki.se (I.D.); 3Department of Clinical Science, Intervention, and Technology (CLINTEC), Karolinska Institutet, 141 86 Stockholm, Sweden; 4Endocrine and Diabetes Unit, Department of Molecular Medicine and Surgery, Karolinska Institutet, 171 77 Stockholm, Sweden; maria.saaf@ki.se; 5Clinical Epidemiology Unit, Department of Medicine, Solna, Karolinska Institutet, 171 77 Stockholm, Sweden

**Keywords:** chronic pancreatitis, bone mineral density, osteoporosis, fracture, PERT

## Abstract

**Introduction:** Chronic pancreatitis (CP) can lead to malnutrition, an established risk factor for low bone mineral density (BMD) and fractures. This study aims to determine the prevalence of low BMD, assess fracture incidence and explore risk factors for fractures in patients with CP. **Patients and methods:** We performed a retrospective analysis of all patients treated for CP at Karolinska University Hospital between January 1999 and December 2020. Electronic medical records were retrieved to assess demographic, laboratory and clinical data. Patients subjected to dual-energy X-ray absorptiometry (DXA) were categorised as either low BMD or normal BMD. We investigated whether the rate of fractures, defined by chart review, differed between these groups using Cox regression, adjusting the model for age, sex and body mass index (BMI). Additional within-group survival analysis was conducted to identify potential risk factors. **Results:** DXA was performed in 23% of patients with definite CP. Some 118 patients were included in the final analysis. Low BMD was present in 63 (53.4%) patients. Mean age at CP diagnosis in the total cohort was 53.1 years and was significantly lower in patients with normal BMD than in patients with low BMD (45.5 vs. 59.8, *p* < 0.001). Significant differences were observed in smoking status and disease aetiology, i.e., a higher proportion of patients with low BMD were current or former smokers, with nicotine or alcohol being a more common cause of CP (*p* < 0.05). Total follow-up time was 898 person-years. Fractures were found in 33 (28.0%) patients: in 5 of 55 patients (16.7%) with normal DXA and in 28 of 63 patients (44.4%) with low BMD (adjusted hazard ratio = 3.4, 95% confidence interval (CI) = 1.2–9.6). Patients with at least 3 months of consecutive pancreatic enzyme replacement therapy (PERT) or vitamin D treatment had a longer median time to fracture after CP diagnosis. **Conclusion:** DXA was only performed in 23% of patients with definite CP in this study, indicating a low adherence to current European guidelines. A low BMD was found in 53.4% of patients with CP, and 44% of the patients with a low BMD experienced a fracture during follow-up. Moreover, the fracture rate in patients with low BMD increased compared to those with normal BMD.

## 1. Introduction

Chronic pancreatitis (CP) can severely affect quality of life and precipitate life-threatening long-term sequelae (e.g., exocrine and endocrine insufficiency, recurrent inflammation, persistent pain and malabsorption), including vitamin and trace mineral deficiencies. Diabetes mellitus (DM), pancreatic exocrine insufficiency (PEI), anorexia secondary to abdominal pain, nausea and vomiting, alcohol, smoking or other substance abuse and poor physical activity levels may all contribute to malnutrition in patients with CP [1].

Osteoporosis is a systemic skeletal disease characterised by low bone mass and microarchitectural deterioration of bone tissue, with a consequent increase in bone fragility and susceptibility to fracture [2]. Based on a meta-analysis, it has been estimated that nearly a quarter of patients with CP have osteoporosis and almost two thirds have either osteoporosis or osteopenia, making these conditions underappreciated sources of morbidity in patients with CP [3]. Current European guidelines recommend preventive measures in all patients diagnosed with CP, with regular quantitative assessment of bone mineral density (BMD) by dual-energy X-ray absorptiometry (DXA) every 2 years in patients with confirmed low BMD. However, bone health assessments are rarely performed in routine clinical practice [1,4,5]. Studies investigating the clinical significance of low-trauma fractures in CP patients estimated a 10-year fracture prevalence of 4.8%, comparable with that of “high-risk” digestive diseases such as liver cirrhosis, inflammatory bowel disease, celiac disease and patients after gastrectomy [6]. Most studies on this topic have been limited by heterogeneous data and small sample sizes not amenable to subgroup analysis [1,3]. This study aims to fill a gap in the literature concerning fracture incidence in CP patients treated at a high-volume European tertiary centre. Therefore, we sought to determine the prevalence of low BMD, fracture incidence and fracture risk factors in patients with CP. In addition, we explored previously suggested contributory factors influencing fracture risk, including treatment with proton-pump inhibitors (PPI), metformin, steroids, opioids or supplementation with vitamin D or pancreatic enzyme replacement therapy (PERT).

## 2. Patients and Methods

### 2.1. Study Population 

In this study, with retrospectively collected data, we first identified all patients with an international classification of diseases (ICD)-based code for CP at the Karolinska University Hospital between January 1999 and December 2020 as eligible for inclusion. Thus, patients could have a first diagnosis of CP before they visited our institution. We excluded patients without a Swedish personal identification number (a 12-digit number issued by the Swedish Tax Agency as a part of the population register used in Sweden), patients with missing or insufficient data in medical charts related to this study, patients without permanent residency in Stockholm County, patients in whom DXA had not been performed and patients with non-definitive CP diagnosis. Baseline data were collected at the first date of diagnosis at our institution. In addition, data were obtained from journal entries available from primary, secondary and tertiary care centres in Stockholm County.

### 2.2. Definitions 

Aetiology of CP was determined according to the M-ANNHEIM classification system, and only patients with definite CP were included in one or more of the following groups: alcohol, nicotine, nutritional factors, hereditary factors, efferent duct factors, immunological and miscellaneous/other [7]. Definite CP was diagnosed by imaging (computed tomography, magnetic resonance imaging or both) with one or more of the following criteria: (a) pancreatic calcifications, (b) moderate or marked ductal lesions, (c) marked and persistent exocrine insufficiency defined as pancreatic steatorrhea markedly reduced by enzyme supplementation or (d) typical histology of an adequate histological specimen. Patients subjected to DXA were grouped into categories of low BMD or normal BMD based on the T-score for BMD assessed by DXA on the femoral neck or lumbar spine, with a value of 1 standard deviation (SD) below that of a healthy 30-year-old of the same sex being indicative of low BMD. In addition, patients with low BMD could be further subdivided into categories of osteoporosis (T-score below or equal to −2.5 SD) or osteopenia (T-score between −1.0 and −2.5 SD). For patients in which only the Z-score had been reported, two standard deviations below normal value were used to define low BMD [5]. Baseline was defined at CP diagnosis. While patients were included if they had received an ICD-based diagnosis code at Karolinska University Hospital between January 1999 and December 2020, the initial diagnosis could have been made earlier in primary care facilities or tertiary centres outside Stockholm. Our primary outcome was defined as time to the first of any fracture following CP diagnosis registered in the electronic medical system available to primary, secondary and tertiary care providers in Stockholm County. Only radiologically confirmed fractures diagnosed in primary, secondary or tertiary care centres in Stockholm County were included. Baseline data were collected concerning patients’ first visit to the Pancreas Outpatient Clinic at the Department of Upper Abdominal Diseases at Karolinska University Hospital. Patients were followed until the first fracture diagnosis was recorded in the electronic medical system or death occurred. Fracture incidence was analysed for BMD after adjusting for patients’ age at baseline, sex and body mass index (BMI). Additional data included the anatomical distribution of fractures, previous diagnosis of DM or PEI at time of CP diagnosis, smoking status (current, never, former) and alcohol consumption (never, ever). High alcohol consumption was defined as a daily alcohol intake of ≥80 g per day (ref number 7). Diagnosis of PEI was based on the concentration of faecal elastase-1 (levels <200 μg/g were considered pathological). Information on DM was obtained from patients’ medical charts. Exploratory analysis was used to investigate whether treatment with drugs that might affect fracture risk was associated with a reduced rate of fractures. Patients were defined as being on treatment if they were on stable maintenance therapy for at least 90 days after baseline with any of the following treatments: PERT, PPI, opioids, oral steroid treatment, vitamin D supplements and metformin.

### 2.3. Statistics 

Categorical variables were compared using Pearson’s chi-square test or Fisher’s exact test as appropriate and reported as percentages and frequencies. Continuous variables, reported as means and tested for significance with a Student’s *t* test, were analysed for normality using the Shapiro–Wilk test. The risk for the primary outcome, i.e., first incident fracture after baseline, was compared between patients with normal BMD and patients with low BMD using Cox regression. Two models were considered, one crude model and one adjusted for age, sex and BMI. Age and BMI were used as continuous variables in the adjusted model. For all regression analyses, results were reported as hazard ratios (HR) and 95% confidence intervals (CI). The Kaplan–Meier method was used to estimate the cumulative incidence of fractures in all patients regarding BMD and drug therapy, and then compared using the log-rank (Mantel–Cox) test. Patients were censored at death. All statistical tests were two-sided and a *p* value of <0.05 was considered statistically significant. All analyses were performed using R Statistical Software (version 4.0.3; R Foundation for Statistical Computing, Vienna, Austria).

### 2.4. Ethics 

The study was approved by the Regional Ethics Committee (Swedish: Regionala Etikprövningsnämnden) in Stockholm, Dnr: 2020-02209. The committee waived the requirement for individual informed patient consent because of the retrospective nature of the study.

## 3. Results

Some 1055 patients diagnosed with CP were eligible for inclusion and 118 were included in the final analysis (a flowchart of patients is presented in Figure 1). In patients with a definitive diagnosis not subjected to DXA, 29% (96/329) had suffered from fractures. This percent is on par with the proportion having suffered fractures in the group of patients who had received DXA, where 28% (33/118) had suffered from fractures. Of the 33 fractured patients subjected to DXA, 17 had performed DXA between CP diagnosis and fracture (16 had DXA following fracture incident). Baseline characteristics for patients in the DXA group at the time of CP diagnosis are presented in Table 1. DXA for assessing BMD with T-score and Z-score was performed at any time point to CP diagnosis and first fracture. Low BMD was present in 63 (53.4%) patients. Total follow-up time was 897.6 person-years (446.2 person-years in patients with low BMD and 451.4 person-years in patients with normal BMD). Mean age at CP diagnosis in the total cohort was 53.1 years and was significantly lower in patients with normal BMD than in patients with low BMD (45.5 vs. 59.8, *p* < 0.001). In addition, females were more prevalent in patients with low BMD than in patients with normal BMD (50.7% vs. 30.9%, *p* < 0.05). Significant differences were also observed in smoking status and CP aetiology. More specifically, a higher proportion of patients with low BMD were current or former smokers (*p* < 0.05), with nicotine or alcohol being a more common cause of the disease (*p* < 0.05).

Some 33 (28.0%) patients suffered from fractures following CP diagnosis. The overall fracture rate between the groups is listed in Table 2. Fractures were more prevalent in patients with low BMD (28/63) compared to patients with normal BMD (5/55). Stratifying patients with low BMD into categories of osteopenia (T-score between –1 and −2.5 SD) or osteoporosis (T-score below –2.5 SD), a higher cumulative incidence of fractures was evident in osteoporotic patients (60% and 30.3% of patients, respectively). Similarly, fracture rate was significantly higher in patients with low BMD than in patients with normal BMD (crude HR = 5.5; 95%CI = 2.1–14.2). Adjusted for sex, age and BMI, the fracture rate remained significantly increased in osteoporotic patients (aHR = 5.5; 95%CI = 1.9–15.8).

The anatomical distribution of fracture locations is given in Table 3 and illustrated in Figure 2. Fractures were evenly distributed among body regions, with hip, vertebra and wrist fractures being more prominent overall in patients with low and normal BMD.

Median time to fracture for patients with low BMD was 9.2 years. Kaplan–Meier curves of cumulative fracture events in patients with normal BMD vs. patients with low BMD are detailed in Figure 3.

Further exploration of the influence of drug therapy before or concomitant with fracture incident on time to first fracture after CP diagnosis is summarised in Appendix A and Appendix A. While the low number of patients not treated with PERT needs to be considered, Kaplan–Meier estimates of median time to first fracture suggest a possible difference between patients treated with PERT supplements and untreated patients. Estimated median time to first fracture was 19.7 years (104 patients, 25 fractures) for PERT-treated patients compared to 6.7 years (14 patients, 8 fractures) to first fracture in PERT-untreated patients. A similar trend can be observed in patients treated with vitamin D supplements, of whom treated patients had a longer median time to first fracture. Estimated median time to first fracture was 12.2 years (48 patients, 19 fractures) for vitamin D-untreated patients compared to 31.7 years (70 patients, 14 fractures) for vitamin D-treated patients. No significant difference could be identified when assessing the influence of steroid, PPI, opioid or metformin treatments.

## 4. Discussion

Patients with chronic pancreatitis are at risk of osteoporosis and fracture because of numerous factors such as deteriorating pancreatic exocrine function, maldigestion and malabsorption of nutrients (especially fat-soluble vitamins and micronutrients), chronic systemic inflammation, abnormal bone turnover and, for some patients, ongoing alcohol excess and smoking [8]. We found low BMD in 53.4% (osteopenia 28.0% and osteoporosis 25.4%) of patients during a mean follow-up of 7.6 years after diagnosis of CP (897.6 person-years), which is lower compared to studies from Ireland [8,9], Germany [10], USA [11] and India [12], but higher than experiences from the Czech Republic [13] (Table 4). A systematic review and meta-analysis that included 10 studies and 513 patients with CP showed a pooled prevalence rate for osteoporosis or osteopenia of 65% (23.4% prevalence for osteoporosis and 39.8% for osteopenia) [3]. However, most of the studies were heterogeneous and not amenable to subgroup analysis, with bias due to small sample size and selection bias leading to wide confidence intervals and statistical uncertainty (using the data on DXA after the fracture is a major selection bias in these kinds of studies) [1,3]. Furthermore, systematic reviews have shown clinical and methodological diversity regarding patient type, disease severity and assessment methods. Hence, any conclusions must be treated with caution [3].

Our study found fracture incidence was 44.4% (*n* = 28) in patients with CP and low BMD; the percent was only 16.7% (*n* = 5) in CP patients with normal BMD. Osteoporosis is a strong risk factor for fractures and there exists a knowledge gap on this topic in patients with CP [3]. In one noteworthy clinical study, Tignor et al. estimated the prevalence of fractures in CP compared to several gastrointestinal diseases. The authors reported a fracture rate of 4.8% in CP, which can be compared to 1.1% in controls, 3.0% in Crohn’s disease, 4.8% in liver cirrhosis and 5.0% in celiac disease [6]. A higher-than-average fracture rate was observed in CP patients compared to population-based controls in a study from Denmark [14]. In that study, the authors reported a higher relative risk of fractures in younger patients with liver cirrhosis and CP, leading to the suggestion that bone loss can be an early component in the disease course. This finding would also corroborate Joshi et al. They demonstrated significantly lower BMD scores at the spine, hip and forearm in young patients with tropical pancreatitis (juvenile form of chronic calcific non-alcoholic pancreatitis, seen almost exclusively in the developing countries of the tropical world) [15]. The fracture rate in CP patients identified in our study was determined over a mean follow-up of 7.6 years and included all fracture incidences in the patient cohort. In contrast, Tignor et al. only included vertebral, hip or wrist fractures, and excluded patients with multiple gastrointestinal diagnoses. We included any first fracture in all patients being treated at our clinic. However, as DXA is a prerequisite for inclusion, this predisposes our cohort to another selection bias for patients with a higher risk of fractures. This concern was partly considered when patients with prior fractures were excluded from analysis. A large proportion of patients had suffered a fracture in the period between diagnosis of CP and DXA scan (16/33), inclining them to receive full osteoporotic work, which will have skewed our final cohort. Median time from CP diagnosis to DXA was similar between patients with low BMD and patients with normal BMD. Regardless, our data on the prevalence of fractures are higher and should be evaluated in other large cohorts of patients with CP, given that it is an ultimate clinical outcome in low BMD.

For patients suffering fractures, treatment with three or more consecutive months with either vitamin D or PERT at any time point before fracture incidence suggested a significantly longer median time to first fracture for all patients. This finding is a new and exciting outcome of our study. Neither previous meta-analyses nor the HaPanEU European guidelines on CP have reviewed any published data on the effect of PERT on fracture risk. Given the dysregulated bone mineral metabolism that follows from severe malnutrition and trace mineral deficiencies, it seems reasonable that ameliorating poor uptake in vitamin and trace minerals could potentially improve BMD. However, until this can be studied in a prospective cohort or register-based cohort, it will remain a significant clinical knowledge gap [1,3].

The mean age in our cohort was 53 years and was significantly lower in patients with normal DXA compared to the low BMD group. Patients in most of the previous studies were younger (Table 4). Due to normal age-related bone loss, it would be expected that older patients (especially females) would have lower BMD. Previous meta-analyses have shown that osteoporosis is prevalent in relatively young patients with CP and no apparent association between age and BMD has been observed [3]. The young age of patients in studies from India can be partially explained by tropical CP, a condition associated with early clinical onset and malnutrition [15].

Females in our study were more prevalent in the low BMD group than in those with normal DXA results, which is another difference from other studies. Surprisingly, no meta-analysis has identified a relationship between sex and low BMD in patients with CP, despite the fact that female sex (especially postmenopausal) is a known risk factor for low BMD [15].

DXA was performed in only 23% of our patients with definite CP, showing low adherence to HaPanEU European CP guidelines. The guidelines recommend DXA every 2 years in patients with CP and low BMD [1].

To our knowledge, this is so far the largest single-centre study on a well-defined cohort of patients with clear differentiation between definite and probable CP and inclusion of patients with various aetiologies of CP, thus representing the treatment reality of CP patients in a high-volume tertiary care centre. The long follow-up is an added strength of the study, as information on treatment and fractures fills the knowledge gaps on this important clinical entity. The major limitation of the study is its retrospective nature. Another important limitation is the lack of data on dietary intake and biomarkers of systemic inflammation. Due to the retrospective analysis, it was not possible to obtain data on DXA on the same date as CP diagnosis. This analysis is another limitation of our study, as well as possible selection bias due to the risk of confounding with patients with comorbidities. Finally, because of the low number of events (only five fractures in the normal BMD group and fourteen patients given PERT), the clinical significance of the influence of these drugs on reducing fracture risk is only suggestive. Our results can be used to inform power calculations for future studies on this topic, which could also include serial DXA with the whole-body composition and information on sarcopenia to evaluate the recently proposed “osteosarcopenia concept” [16], detailed dietary intake and follow-up of all anthropometric parameters, as well as bone resorption and bone modelling factors (biomarkers).

**Table 4 nutrients-13-02386-t004:** Studies published on osteopathy in chronic pancreatitis.

Author	Year	Country	N	Age	Sex	Aetiology/Results	Comments
Moran [17]	1997	Argentina	14	56	Not mentioned	Alcohol: 71.4% Idiopathic 28.6% Osteopathy: 92.8% Osteopenia: 71.4% Osteoporosis: 21.4%	All included patients with severe PEI.
Haaber [18]	2000	Denmark	58	54	55.2% male	Alcohol: 79% 56% of patients in the group without PEI and 69% in the group with PEI had Z-scores of the BMD < –1	Conclusions: patients with CP, particularly patients with advanced disease and steatorrhea, are at risk of developing significant bone loss
Mann [19]	2003	Germany	42	51.1	All males	Not described	Conclusion: connection between the inflammatory destruction of the pancreas (Cambridge classification), exocrine pancreatic insufficiency (faecal elastase 1), altered levels of vitamin D metabolites and loss of skeletal mass
Dujsikova [13]	2008	Czech Republic	73	46.6	76.7% males	Osteopathy: 39%. Osteopenia: 26%, Osteoporosis: 5%, Osteomalacia: 8%	CP diagnosed with EUS and defined as mild, moderate and severe (more than half of the patients had mild CP)
Sudeep [20]	2011	India	31	35.8	All males	Tropical pancreatitis: 65% Idiopathic: 35% 29% had a T-score of less than −2.5	Conclusion: Patients with chronic pancreatitis and a T-score of < −2.5 had a significantly lower BMI. No correlation was found between 25(OH)D levels and BMD
Joshi [15]	2011	India	72 (DXA in 60)	31	53% males	All patients with tropical calcific pancreatitis	Conclusion: despite their young age, patients with tropical calcific pancreatitis have significantly low BMD
Duggan [9]	2012	Ireland	53	48.7	75.5% males	Alcohol was the cause of disease in 38.7% of patients. (just over 6% of both patients and controls were never drinkers). Osteopathy: 73.6% Osteoporosis: 34% Osteopenia: 39.6%	Conclusion: a third of the patients with CP had osteoporosis, which was more than triple the rate in the matched control group
Sikkens [21]	2013	The Netherlands	40	52	57% males	Alcohol: 50% Idiopathic: 43% Other: 7% Osteoporosis: 10% Osteopenia: 45%	Deficiencies of fat-soluble vitamins and a decreased BMD are frequently present in CP, even in exocrine-sufficient patients. Consequently, all patients with CP should be routinely screened for fat-soluble vitamin deficiencies and a decreased BMD
Duggan [8]	2014	Ireland	29	44.3	58.6% male	Alcohol: 62.1% Idiopathic: 27.6% Other: 10.3% Osteoporosis: 31% Osteopenia: 44.8%	Conclusion: both bone formation and bone resorption were raised in patients with CP compared to controls. This finding indicates that bone turnover was elevated in CP. Those with alcohol-induced disease did not have lower BMD than those with CP of other aetiologies
Prabhakaran [12]	2014	India	103	38.6	All males	Alcohol: 70%, Idiopathic: 29.1%; (one patient had post-traumatic chronic Pancreatitis) Osteoporosis: 30.1%, Osteopenia: 39.8%	Conclusion: most patients with both alcoholic and idiopathic had low BMD and the frequency of bone changes was similar between calcific and non-calcific groups, diabetics and nondiabetics, patients with and without a history of steatorrhea, patients with and without vitamin D deficiency and across different pancreatitis severity groups
Min [11]	2014	USA	91	48.6	62.6% females	Toxic/metabolic: 59.3% Idiopathic: 18.6% Hereditary: 14.3% Autoimmune: 5.5% Osteopenia: 46.7% Osteoporosis: 22.2%	Conclusion: There is a high prevalence of fat-soluble vitamin deficiencies, osteopathy and malnutrition in CP patients, which is underestimated due to the lack of effective diagnosis and suboptimal therapies for EPI
Haas [10]	2015	Germany	15	45.2	all males	Alcohol: 72% Osteoporosis: 16% Osteopenia: 76%	Conclusion: no correlation between bone metabolism and elastase
Stigliano [22]	2018	European multicentric	211	60	67% males	Alcoholic: 43% Idiopathic: 19% Hereditary: 4% Obstructive: 5.7% Osteopenia: 42% Osteoporosis: 22%	Conclusion: vitamin K deficiency is the only factor associated with osteoporosis in male patients
Present study	2021	Sweden	118	53.1	58.5% males	Alcohol and smoking: 33.9% Smoking only: 11% Alcohol only: 5.9% Hereditary: 11.8% Immunological: 14.4% Efferent duct factors: 9.3% Low BMD: 53.4%	Conclusion: Low BMD was found in 58% of patients with CP with a high prevalence of fractures (53%). Most of the fractures occurred in patients with low BMD but also occurred in patients with normal DXA. Previous treatment with either vitamin D or PERT demonstrated a significantly lower risk for fractures in all patient groups

N = number of patients included in the study; CP = chronic pancreatitis; PEI = pancreatic exocrine insufficiency; BMD = bone mineral density; DXA = dual-energy X-ray absorptiometry; PERT = pancreatic enzyme replacement therapy.

## 5. Conclusions

In this study, DXA was performed in only 23% of patients with definite CP, showing a low adherence to HaPanEU European guidelines on CP. A low BMD was seen in 53.4% of patients with CP, and 44% of the patients with a low BMD experienced a fracture during follow-up. Moreover, the fracture rate in patients with low BMD was increased compared to patients with normal BMD. Considering the frequency and clinical importance of fractures in CP and the potentially positive effect of lifestyle modification, environmental factors and secondary prevention, it is worthwhile to create patient and physician awareness of this clinical entity. As PEI has been associated with a lower BMD, it remains imperative to conduct prospective studies on the role of PERT as a cornerstone in the treatment of PEI.

## Figures and Tables

**Figure 1 nutrients-13-02386-f001:**
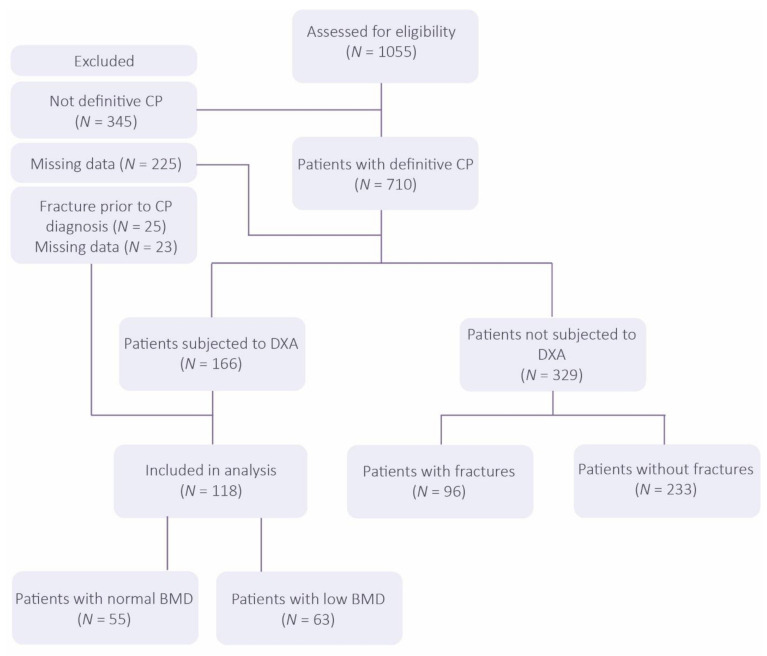
Flow chart of patient selection (CP = chronic pancreatitis; BMD = bone mineral density; DXA = dual-energy X-ray absorptiometry).

**Figure 2 nutrients-13-02386-f002:**
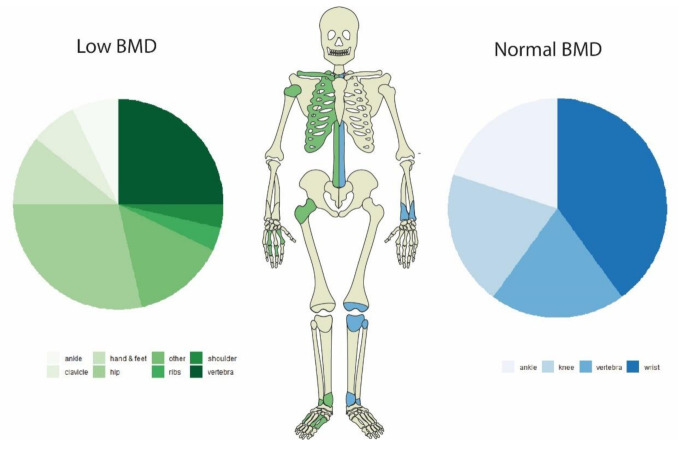
Distribution of fractures by bone mineral density (BMD). Fractures in both groups were evenly distributed among body regions, with hip, vertebra and wrist fractures being more prominent overall.

**Figure 3 nutrients-13-02386-f003:**
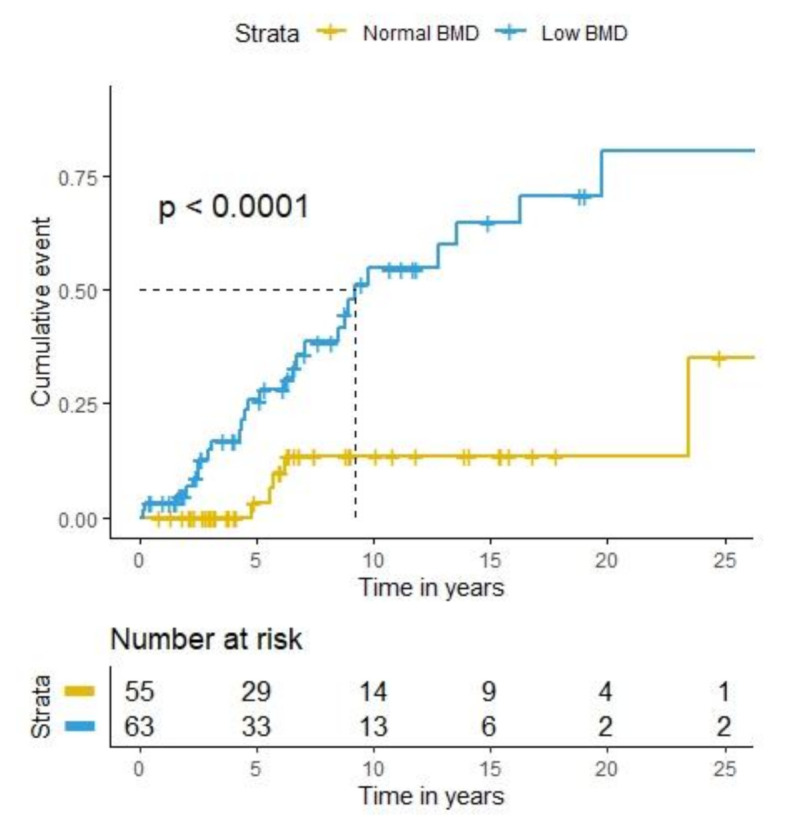
Kaplan–Meier curve of estimated cumulative fracture events for patients stratified by bone mineral density (normal BMD vs. low BMD). BMD = bone mineral density.

**Table 1 nutrients-13-02386-t001:** Baseline patient characteristics.

	Total (*n* = 118)	Low BMD (*n* = 63)	Normal BMD (*n* = 55)	*p*-Value
Female sex, *n* (%)	49 (41.5)	32 (50.7)	17 (30.9)	<0.05
Follow-up (years), mean (SD)	7.6 (7.5)	7.1 (6.8)	8.2 (8.1)	0.424
Age at CP diagnosis (years), mean (SD)	53.1 (16.3)	59.8 (11.8)	45.5 (17.5)	<0.001
Diabetes at diagnosis, *n* (%)	28 (23.7)	16 (25.4)	12 (21.8)	0.774
Age group, *n* (%)	-	-	-	-
- <45 years	35 (29.6)	6 (9.5)	29 (52.7)	-
- 45–65 years	51 (43.2)	35 (55.6)	16 (29.0)	-
- ≥65 years	32 (27.1)	22 (35.9)	10 (18.1)	<0.005
BMI, mean (SD)	23.9 (4.4)	23.06 (4.05)	24.9 (4.6)	<0.05
- BMI <20.0, *n* (%)	25 (21.2)	15 (23.8)	10 (1.1)	-
- 20.1 < BMI < 25.0, *n* (%)	50 (42.4)	33 (52.4)	17 (30.9)	-
- 25.1 < BMI < 30.0, *n* (%)	30 (25.4)	10 (15.9)	20 (36.3)	-
- 30.1 < BMI, *n* (%)	13 (11.0)	5 (7.9)	8 (14.5)	<0.05
Smoking status	-	-	-	-
- Current smoker, *n* (%)	42 (35.6)	28 (44.4)	14 (25.5)	-
- Former smoker, *n* (%)	34 (28.8)	19 (30.2)	15 (27.2)	-
- Never smoker, *n* (%)	42 (35.6)	16 (25.4)	26 (47.2)	<0.05
Alcohol status				
- Non-drinker	64 (54.7)	28 (44.4)	36 (66.7)	
- Drinker	53 (45.2)	35 (55.6)	18 (33.3)	<0.01
- Data not available	1	0	1	
Aetiology of CP	-	-	-	-
- Alcohol and nicotine, *n* (%)	40 (33.9)	27 (46.6)	13 (26.5)	-
- Nicotine, *n* (%)	13 (11.0)	9 (15.5)	4 (8.1)	-
- Alcohol, *n* (%)	7 (5.9)	4 (6.9)	3 (6.1)	-
- Hereditary, *n* (%)	14 (11.8)	3 (5.2)	11 (22.4)	-
- Immunological, *n* (%)	17 (14.4)	9 (15.5)	8 (16.3)	-
- Immunological factors and nicotine, *n* (%)	0 (0.0)	0 (0.0)	0 (0.0)	-
- Efferent duct	11 (9.3)	6 (10.3)	5 (10.2)	-
- Misc./Other, *n* (%)	8 (6.7)	3 (5.1)	5 (10.2)	<0.05
- Data not available	8	2	6	-
PEI at diagnosis	-	-	-	-
- No *n* (%)	35 (39.3)	20 (40.0)	15 (38.5)	
- Yes *n* (%)	54 (60.7)	30 (60.0)	24 (61.5)	1.0
- Missing data	29	13	16	-
DXA results, median (quartiles)	-	-	-	-
- T-score hip	−1.47 (−2.36, 0.81)	−2.18 (−2.6, 1.7)	−0.3 (−0.85, 0.2)	<0.005
- T-score lower back	−0.71 (−2.0, 0.55)	−1,55 (−2.78, −0.5)	0.69 (0.1, 1.2)	<0.005
Median time from CP diagnosis to DXA, years (IQR)	2.8 (7.4)	2.8 (8.0)	2.7 (6.5)	0.696

CP = chronic pancreatitis; SD = standard deviation; BMI = body mass index; PEI = pancreatic exocrine insufficiency; DXA = dual-energy X-ray absorptiometry. BMD = bone mineral density.

**Table 2 nutrients-13-02386-t002:** Overall rate of fractures.

	N	Event	>Person-Years	Incidence	Crude HR [CI 95%]	aHR * [CI 95%]
Normal BMD	55	5	451.4	1.1	1.0 [ref.]	1.0 [ref.]
Low BMD	63	28	446.2	6.3	5.5 [2.1, 14.2]	3.4 [1.2, 9.6]
- Osteopenia	33	10	248.6	4.0	3.5 [1.2, 10.4]	2.2 [0.7, 6.8]
- Osteoporosis	30	18	197.6	9.2	7.8 [2.9, 21.01]	5.5 [1.9, 15.8]

N = number of patients; HR = hazard ratio; aHR = adjusted hazard ratio; BMI = body mass index; CP = chronic pancreatitis. Ref. = referent. CI = confidence interval. Time from CP diagnosis to first fracture. Incidence per 100 person-years. * Adjusted for sex, age and BMI.

**Table 3 nutrients-13-02386-t003:** Fracture incidence by anatomic region.

	Low BMD			Normal BMD		
	Event	Person-Years	Incidence	Event	Person-Years	Incidence
Hip	8	446.2	1.8	0	451.4	0.0
Vertebrae	7	446.2	1.6	1	451.4	0.22
Rib	1	446.2	0.22	0	451.4	0
Ankle	2	446.2	0.45	1	451.4	0.22
Shoulder	1	446.2	0.22	0	451.4	0
Wrist	0	446.2	0.0	2	451.4	0.44
Clavicle	2	446.2	0.45	0	451.4	0.0
Hand & feet	3	446.2	0.67	0	451.4	0.0
Knee	1	446.2	0.22	1	451.4	0.22
Other	3	446.2	0.67	0	451.4	0.0

N = number of patients. BMD = bone mineral density. Incidence per 100 person-years.

## Data Availability

Our dataset contains sensible data which may contain private informa- tion about the patients treated at out clinic. The dataset can therefore not be made available to the public. However, the data used in our study can be provided upon request.

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
