# Peer review of "Low Bone Mineral Density and Risk for Osteoporotic Fractures in Patients with Chronic Pancreatitis"

_nutrients, 2021, doi:10.3390/nu13072386_

Round 1

Reviewer 1 Report

In this manuscript, authors performed the clinical study of CP-related low BMD which may result in fracture. This study seems to be interesting and important for the clinician dealing with CP patients. However, the manuscript lacks direct evidence and explanation why CP can induce low BMD.

1) Can low BMD be induced by only CP ? 

2) Low BMD can be induced by various factors. Authors must explain how CP-induced low BMD is different with low BMD induced by another factors. 

3) Is fracture incidence directly related to CP-induced low BMD? 

Author Response

Answer to reviewers:

Thank you very much for your valuable comments!

Please find enclosed answers point by point.

Reviewer 1:

In this manuscript, authors performed the clinical study of CP-related low BMD which may result in fracture. This study seems to be interesting and important for the clinician dealing with CP patients. However, the manuscript lacks direct evidence and explanation why CP can induce low BMD.

1) Can low BMD be induced by only CP? Low BMD can be induced by various factors. Authors must explain how CP-induced low BMD is different with low BMD induced by another factors.

Authors’ answer:

We agree with your comment. We made changes and added explanation at the beginning of discussion section.

“Patients with chronic pancreatitis are at risk of osteoporosis and fracture because of numerous factors such as deteriorating pancreatic exocrine function, maldigestion and malabsorption of nutrients (especially fat-soluble vitamins and micronutrients), chronic systemic inflammation, abnormal bone turnover and for some patients ongoing alcohol excess and smoking.”

3) Is fracture incidence directly related to CP-induced low BMD?

Authors’ answer:

Unfortunately, retrospective nature of analysis is a limitation of the manus, and causality cannot be proven in our retrospective cohort (presence of other factors is possible). However, statistics showed significant association that is presented in table 2 (HR based on BMD status).

Reviewer 2 Report

The paper entitled "Low bone mineral density and risk for osteoporotic fractures in patients with chronic pancreatitis" by Miroslav Vujasinovic and colleagues presents a retrospective analysis to evaluate the effect of chronic pancreatitis on bone fragility. 
Although the topic is of primary importance for the management of these patients in the long period, the study raises some major concerns that need to be addressed before publication:
-Age, hormones, and smoking are well-known factors affecting BMD. 
It is not clear if these factors improve the fracture risk in CP patients?
-The higher fracture rate in the low BMD group is not surprising. 
The overall risk for each group should be compared with age- sex- and BMD-matched non-CP subjects
-Kaplan-Meier analysis of CP patients treated with PERT as reported in Supplementary material is the most interesting and innovative part of the study and should be moved in the main text.
-Page 4: part of the sentence is hidden by figure 1

Author Response

Answer to reviewers:

Thank you very much for your valuable comments!

Please find enclosed answers point by point.

Reviewer 2:

The paper entitled "Low bone mineral density and risk for osteoporotic fractures in patients with chronic pancreatitis" by Miroslav Vujasinovic and colleagues presents a retrospective analysis to evaluate the effect of chronic pancreatitis on bone fragility. Although the topic is of primary importance for the management of these patients in the long period, the study raises some major concerns that need to be addressed before publication:

-Age, hormones, and smoking are well-known factors affecting BMD. It is not clear if these factors improve the fracture risk in CP patients? –The higher fracture rate in the low BMD group is not surprising. The overall risk for each group should be compared with age- sex- and BMD-matched non-CP subjects

Authors’ answer:

We agree with your comment. Unfortunately, due to retrospective nature of our study, we were not able to get individual data on menopause, HRT-treatment, testosterone, and other important hormone-related information. This question is very important, and all these factors will be included in the future work of our group - we are planning prospective clinical study (hopefully it will be Scandinavian multicentric study because the number of patient in single centers is low), as well as register-based study for Sweden.

-Kaplan-Meier analysis of CP patients treated with PERT as reported in Supplementary material is the most interesting and innovative part of the study and should be moved in the main text.

Authors’ answer:

We agree with your comment of importance of this part of manus. It was included in the original manuscript (during the manus writing) but after the discussion in the team, we decided to add it in the supplement. Because of the low number of events (only 5 fractures in the low BMD group and 14 patients given PERT), the clinical significance of the influence of these drugs on reducing fracture risk is only suggestive. Therefore, we decided to present this result carefully: mentioning in the results section and discussion section but avoiding strong conclusion. Therefore we presented figure in a supplement and we hope that reviewer will accept our explanation.

-Page 4: part of the sentence is hidden by figure 1

Authors’ answer:

Thank you for the comment – we will ask editorial what the problem was because in original document it looks good.

Reviewer 3 Report

This is a nice work, however not free of some limitations.

Major comments:

  1. Methods are not precisely described : no information is given whether BMD was measured using the same type of instrument. To my knowledge there are two main manufacturers of DXA instruments and their precision may differ. Regarding the diagnosis of diabetes : the HbA1c criterion was recommended only in 2011 by NACB and the patients data covered those diagnosed with CP between 2009-2020. Therefore it would be of importance to mention this fact.  The authors should also correct the cut-off  for glucose during 2hrs OGTT which obviously is >11.1 mmol/L  not 1.1 mmol/L. Another unfortunate/curious  description refers to fecal elastase-1  which was "measured in feces".
  2. Data on vitamin D treatment effects on fracture prevalence also need a detailed commentary as the patients for sure were given different formulas and different doses. It would also be interesting to know which kind of vitamin D (D2 or D3) was mostly given to the patients.
  3. Conclusions should be rewritten as in the presented form they repeat the obtained results.  I suggest that the last section  of Discussion better fits as conclusions than the one presented in the submitted article.

Author Response

Answer to reviewers:

Thank you very much for your valuable comments!

Please find enclosed answers point by point.

Reviewer 3:

This is a nice work, however not free of some limitations. Major comments:

1.Methods are not precisely described: no information is given whether BMD was measured using the same type of instrument. To my knowledge there are two main manufacturers of DXA instruments, and their precision may differ. Regarding the diagnosis of diabetes: the HbA1c criterion was recommended only in 2011 by NACB and the patients data covered those diagnosed with CP between 2009-2020. Therefore, it would be of importance to mention this fact. The authors should also correct the cut-off for glucose during 2hrs OGTT which obviously is >11.1 mmol/L not 1.1 mmol/L. Another unfortunate/curious description refers to fecal elastase-1 which was "measured in feces".

Authors’ answer:

We agree with your comments. Unfortunately, due to retrospective analysis of data, we were not able to find detailed information on instrument and manufacturer for each patient. We checked articles from Karolinska osteoporosis group published at PubMed during the same period and all studies were performed with Lunar and Hologic instrument. We agree that these instruments are not the same but information from DXA report important for our study (T-score) should not be significantly different. Also, we were analyzing data from all patients’ medical charts, including primary and secondary health care as well as our DXA results, therefore, we decided not to mention info on instruments in manuscript.

Information on diabetes was obtained from patients’ medical charts. The text has been updated accordingly and information about diagnostic criteria for diabetes excluded since this is not the focus of the present manuscript.

We corrected curious description of fecal elastase (we are thankful for this comment and apologize for the mistake).

2.Data on vitamin D treatment effects on fracture prevalence also need a detailed commentary as the patients for sure were given different formulas and different doses. It would also be interesting to know which kind of vitamin D (D2 or D3) was mostly given to the patients.

Authors’ answer:

This is relevant comment and we agree with it. All patients received cholecalciferol (D3) but have not gathered data on range or median dose. This is also one of the limitations or retrospective analysis and additional reason to avoid strong conclusion regarding the effect of drugs (we added figures regarding this topic in supplement data and commented significance of result in discussion section of manuscript).

3.Conclusions should be rewritten as in the presented form they repeat the obtained results. I suggest that the last section of Discussion better fits as conclusions than the one presented in the submitted article.

Authors’ answer:

We agree with your comments and we made changes in the text accordingly.

Reviewer 4 Report

1.A clinical team of ten authors addresses the incidence of low bone mineral density and the risk of osteoporotic fractures in chronic pancreatitis in an article that is well considered, appropriately illustrated and clearly written, although a less emphatically personal style is preferable, i.e. the personal pronoun “we” is over-used throughout.

2.The current literature on the topic is opened to criticism as it is apparently confined to male subjects  in all 10 previous studies totalling 513 patients.  The authors are encouraged to make more of this point they have raised, especially since trabecular patterns of age-related bone loss differ fundamentally between the sexes and significantly their own patients are female, thereby helping to restore the balance.

3.The investigation is commendably self-critical ( e.g., retrospective in nature; lack of biomarkers; presence of comorbidities, etc) in its application to 118 patients (mean age at diagnosis 53yr) of which about half had a lower BMD and its comparison with other reports is unavoidably limited by inherent methodological diversity and frequently small sample size.

4.An osteopenic outcome of the associated malnutrition might be expected, however, a strong case is made for more routine DXA in monitoring skeletal status in CP in the longer term and especially with regard to treatment. PERT and vitamin D therapy is considered (including a future anticipated complementary prospective cohort study) and is an aspect that seems to warrant inclusion in the Abstract. Related is the question of whether common drugs for the prevention of osteoporosis, such as the bisphosphonates, have a potential role in CP treatment, and if not, why not?

5.A minor point in Table 1 is occasional misalignment.

Author Response

Answer to reviewers:

Thank you very much for your valuable comments!

Please find enclosed answers point by point.

Reviewer 4:

1.A clinical team of ten authors addresses the incidence of low bone mineral density and the risk of osteoporotic fractures in chronic pancreatitis in an article that is well considered, appropriately illustrated and clearly written, although a less emphatically personal style is preferable, i.e. the personal pronoun “we” is over-used throughout.

Authors’ answer:

The authors thanks for the positive reception of the manuscript and valuable comments on language. We reduced use of pronoun “we”.

2.The current literature on the topic is opened to criticism as it is apparently confined to male subjects in all 10 previous studies totaling 513 patients. The authors are encouraged to make more of this point they have raised, especially since trabecular patterns of age-related bone loss differ fundamentally between the sexes and significantly their own patients are female, thereby helping to restore the balance.

Authors’ answer:

This is excellent question and in in the line with the endocrinologist part of our team. Unfortunately, we are not able to offer more data on this topic in our retrospective study. We do not know if the women are hypoestrogenic or not (very young sick women can become hypogonadal and a few women keep menstruating after 55 years of age). Women have a higher risk of fracture than men due to lower peak bone mass, other bone dimensions, mandatory loss of gonadal hormones (median age 51-52 years, about a year earlier if smokers) with accelerated bone loss. The fracture pattern is often “trabecular fractures” in the early years after menopause (first wrist, thereafter upper arm, vertebral bodies and late in life hip and pelvis). Only prospective study can help us to fill the knowledge gap here (we are hoping on multicentric Scandinavian study due to the low number of patients in single centers).

3.The investigation is commendably self-critical (e.g., retrospective in nature; lack of biomarkers; presence of comorbidities, etc) in its application to 118 patients (mean age at diagnosis 53yr) of which about half had a lower BMD and its comparison with other reports is unavoidably limited by inherent methodological diversity and frequently small sample size. An osteopenic outcome of the associated malnutrition might be expected, however, a strong case is made for more routine DXA in monitoring skeletal status in CP in the longer term and especially regarding treatment. PERT and vitamin D therapy is considered (including a future anticipated complementary prospective cohort study) and is an aspect that seems to warrant inclusion in the Abstract. Related is the question of whether common drugs for the prevention of osteoporosis, such as the bisphosphonates, have a potential role in CP treatment, and if not, why not?

Authors’ answer:

Thank you very much for positive reception and proper interpretation of the most important messages of our work. We wrote following sentence in the abstract: “Patients with at least 3 months of consecutive pancreatic enzyme replacement therapy (PERT) or vitamin D treatment had a longer median time to fracture after CP diagnosis.” During the writing of article, we wanted to emphasize the importance of drugs but after careful interpretation of data we decide to be more careful. Because of the low number of events (only 5 fractures in the low BMD group and 14 patients given PERT), the clinical significance of the influence of these drugs on reducing fracture risk is only suggestive. Therefore, we decided to present this result carefully: mentioning in the results section and discussion section but avoiding strong conclusion.

In younger patients with CP the most important is to correct malnutrition and deficits (like vitamin D, vitamin K and calcium, if needed) and encourage good life style habits (no tobacco, no alcohol, regular exercise, balance training, normal weight). In older patients with low BMD, higher fracture risk or even already manifested osteoporotic fracture, bone specific drugs can also be considered when osteomalacia has been out ruled. Use of bisphosphonates is related to secondary prevention in these patients but (to our best knowledge) not to specific treatment of CP.

5.A minor point in Table 1 is occasional misalignment.

Authors’ answer:

Thank you for the comment. We made corrections appropriately.

Round 2

Reviewer 1 Report

Issues I have addressed are resolved. 

Reviewer 2 Report

Despite not totally convincing replies to reviewer comments, the manuscript can be published on Nutrients.